# Offspring Education and Parents’ Health Inequality in China: Evidence from Spillovers of Education Reform

**DOI:** 10.3390/ijerph19042006

**Published:** 2022-02-11

**Authors:** Youlu Zhang, Li Zhang, Fulian Li, Liqian Deng, Jiaoli Cai, Linyue Yu

**Affiliations:** 1School of Economics and Management, Beijing Jiaotong University, Beijing 100044, China; 18113024@bjtu.edu.cn (Y.Z.); lzhang@bjtu.edu.cn (L.Z.); 18113002@bjtu.edu.cn (L.D.); jiaoli.cai@bjtu.edu.cn (J.C.); 2School of Economics and Management, Shandong Agricultural University, Tai’an 271018, China; lifuliansdau@163.com

**Keywords:** offspring education, parents’ health inequality, university enrollment expansion policy, spillovers, quantile treatment effect, China

## Abstract

In the context of a rapidly aging population, improving the parents’ health outcomes, especially in parents with poorer health, is essential for narrowing elderly health inequality. Using data from the China Health and Retirement Longitudinal Study, we took the university enrollment expansion policy as the instrumental variable and employed the two-stage least square (2SLS) and instrumental variable quantile regression (IVQR) approaches to explore the spillovers of offspring education on the elderly parents’ frailty index. The results show that one additional year of offspring educational attainment was associated with a 0.017 or 4.66% decline in the parents’ frailty index. These spillovers are stronger where parents are cohabiting with their children than when separating (more than 2 times higher). Moreover, there is substantial heterogeneity that is determined by the gender of parents. The spillover on mothers is greater than that on fathers. Further analysis of a cohort of parents with different frailty indexes reveals that the upward spillovers of offspring education on parents’ health are non-linear and non-averaged. The spillovers may diminish as parents own health improves. These spillovers suppress the “Matthew Effect”, which can lead to the further widening of health inequality.

## 1. Introduction

Human life expectancy continues to increase, and the birth rate continues to decline. Many countries are facing the challenges associated with having an inverted pyramid-shaped aging population, especially in developing countries such as China. At present, the aging population is a serious situation in China. The problem of “Getting Old Before Getting Rich” is prominent, arousing concerns among the Chinese government. The aging population in China has led to a significant increase in the dependence on support of the elderly. Given the imperfection of China’s social pension system, the health problems that are experienced by the elderly population, especially in rural areas, create many challenges related to the sustainable development of public health undertakings in China [1,2].

For a long time, education has been shown to create a variety of non-economic benefits, including improved health [3,4,5]. Among the many factors that affect the health of the elderly, education is one of the most important considerations [6]. The life-course perspective proposed by Elder [7] indicates “linked lives” or the interdependence of family members and posits that individual life exists in interdependence. The characteristics of a person’s life may be influenced by other people’s life characteristics, such as education and health. In particular, mounting evidence suggests that the health effects associated with education can be transmitted among different individuals and across generations, such as from parents to children [8,9], thus creating positive externalities. These spillovers from one generation to the other should be considered when evaluating the societal returns to education [10].

However, the intergenerational mobility of education and health should not only work in one direction from parent to child. On the contrary, there are reasons to believe that intergenerational mobility might work in the opposite way as well. The Chinese family system provides social support and assistance to all family members through resource sharing [11]. After the parents reach old age, the resources that are provided by their offspring are the major safeguard of them. For example, well-educated children are better able to reduce financial stress for their parents. Parents may be proud of their children’s success. This is good for protecting the psychological well-being of parents and increases their happiness later in life [12]. Children can invest more time and material resources in the production of their parents’ health and act as their agents in health and long-term care systems [13]. These views have thus aroused our interest in the following questions: Are there spillovers of offspring education on their parents’ health? If these spillovers exist, would they reduce health inequalities among parents of different cohorts? Specifically, there has been a dramatic increase in the education on the younger Chinese cohorts since the implementation of the college expansion policy in 1999, so the possible spillovers caused by this have aroused our interest.

Our article provides causal evidence on the spillovers of offspring education on parents’ health inequality in China by following a similar approach and focusing on college education. Most of the previous research only analyzes one or more of the following: the mortality, cognition, self-rated health, and mental health of parents [12,14,15,16]. These variables can reveal different subjective and objective health [17]. However, health is a multidimensional concept, and mortality or self-reported health do not reflect outcomes across the general spectrum of health. Based on the existing research, this study constructed a comprehensive health index of parents, the frailty index from both subjective and objective health measures, covering multiple aspects including physical, mental, and cognitive health outcomes. The use of this index expands the connotation of a health evaluation index and enriches the related research in this field. Through this research, we will be able to understand the spillovers of offspring education on the general health of elderly parents.

While previous literature has begun to explore the relationship between offspring education and parents’ health, few studies that have explicitly attempted to address questions of causality have assessed whether offspring education plays a role in parents’ health. Due to the endogeneity of education, previous analyses have worried that education is correlated with unobserved factors between offspring and parents, such as innate ability and family background. In addition, healthy elderly people also have more resources that able to invest in their children’s education. Considering that the association between offspring education and parents’ health is not the only imaginable connection, these identification threats become even more important [18]. Therefore, we utilized as quasi-natural experiments the implementation of the university enrollment expansion policy from 1999 across China. The implementation of this policy generated exogenous variation in years of offspring education across both cohorts. To a large extent, this solved the endogeneity problem of education. The results of our research show that when children are able to receive a higher level of education, it leads to a decrease in the frailty index of the Chinese elderly. Additionally, we also examined the gender heterogeneity of these spillovers. Offspring education has been found to have a stronger association with mothers’ health compared to fathers. This result is not entirely consistent with the results of recent findings [16,18]. This reflects the traditional Chinese ideas about gender. On the one hand, men and women are still arranged in different spheres, breadwinner and caregiver, in the elderly labor market. On the other hand, due to physiological features of females, it is very hard for them to escape from the responsibilities of being homemakers and caregivers in the family. As a result, many females provide care to their children when younger and to their grandchildren more when older, especially in agricultural communities. Mothers, who are often regarded as primary caregivers, are more likely to maintain closer contact with their offspring.

This study also expands upon existing work. We further explored the differences in spillovers between cohorts of parents who with different levels of health to better understand the health inequality of elderly parents related to offspring education. The law of diminishing marginal utility indicates that the incremental utility that consumers achieve from continuously increasing consumption units is diminishing. That is to say, we can deduce that the spillovers of offspring education may diminish as parents’ health gradually improves. However, whether the spillovers of offspring education are universal when parents are experiencing different levels of health is still an issue that has not been fully discussed. To address this problem, we utilized instrumental variable quantile regression (IVQR) to analyze the differences in the spillovers at different health quantiles. Our evidence indicates that the spillovers of offspring education are not the same among the elderly with different levels of health. The law of diminishing marginal utility also applies to the spillovers of offspring education. This finding is of great significance for reducing health inequality. The resources that are supplied by one’s children may be of particular importance for the elderly in China because the vast majority of this populations relies on support from their adult children [19]. Our paper covers these issues and provides new evidence from China for reducing social inequalities in health and for evaluating the impact on the health inequality of non-health sector policies.

The rest of the paper is organized as follows. Section 2 outlines the theoretical background. Section 3 briefly presents education reform (university enrollment expansion) in China. Section 4 describes the data and variables in analysis. Section 5 recommends our empirical strategy. Section 6 reports the estimation results. Section 7 provides additional robustness checks. Section 8 concludes and discusses the results of our article.

## 2. Theoretical Background

### 2.1. Offspring Education and Parents’ Health Inequality

Health demand theory holds that education is one of the most important factors that impacts individual health [20]. Subsequent research has also confirmed this view [21,22]. Many types of research have examined the social returns of education among children for their parents. Scholars found that offspring education has a positive impact on parents’ health across multiple dimensions [14,15,23,24]. The increase in schooling creates cross-individual spillovers within the family through the positive externalities of education [25]. Parents benefit human capital gains of health that result from the increase in their children’s schooling [26]. These benefits vary by region and income. In low- and middle-income areas with inadequate public support and social security, such as in Tanzania, South Africa, and Nepal, the spillovers are even stronger [26,27,28].

An increasing number of studies have confirmed that there is gender heterogeneity among parents in the spillovers of offspring education, making generalizations difficult. Some scholars believe that the spillovers of offspring education on mothers’ health outcomes is greater than that on fathers’ [24,28,29]. This is mainly determined by the position that is held by women in the labor market as well as by their role in the family structure. The gender disadvantage in the labor market means that mothers experience a relatively loose attachment to the labor market and are more dependent on support from their children and other family members. Social and cultural norms emphasize as the role of females as homemakers and caregivers. The mother is also responsible for caring for her children and grandchildren as family kin keepers, which occupies a more central role in family life in China compared to fathers. This makes the relationship between mother and her offspring even closer. Other scholars argue that males are more likely to perform manual labor in the labor market and to partake in unhealthy behaviors, such as excessive drinking and smoking. This means that there is more room for improvement in the health outcomes of fathers and they are more sensitive to the spillovers of their offspring education. Therefore, offspring have a potentially greater impact on the health of their fathers than on their mothers [18,30]. However, Jiang [16] believes that there is no gender heterogeneity among parents regarding the spillovers of offspring education. He points out that traditional cultural norms of filial piety create a balanced relationship between generations in East Asian countries such as China. There is no significant difference in the spillovers on health outcomes of fathers or mothers. Thus, this conclusion should be interpreted with caution due to the diversity in generational relations and social cultures across countries.

Meanwhile, we must also consider that different living arrangements may change the spillovers of offspring education. As social modernization continues to improve in China, the traditional extended family, with multiple generations living together, is incompatible with the basic requirements of modern society. The development of modernization will inevitably cause the family structure to modernize as well; that is, there are decreases in family size and changes to the migration process [17]. More and more Chinese children are choosing to live alone after marriage [31]. The burden of supporting the elderly falls more heavily on the one child. As a key component of intergenerational connections, changes in living arrangements may cause changes in the relationship between offspring education and parents’ health [32]. Thoma et al. [32] found that in India, the education of children who live with their parents is significantly related to health improvements in the elderly. It is more possible for well-educated offspring to live with parents and to have more frequent contact with their parents than with less education [29], which help the spillovers have an opportunity to work more effectively. However, Jiang [16] believes that offspring education is an important factor in the low rate of cohabitation between parents and children in China. It is difficult for college-educated offspring to provide direct care for their parents [16]. These make the net impact of living arrangements on the spillovers of offspring education theoretically more ambiguous.

Research is also needed to identify the underlying mechanisms linking offspring education to parents’ health. These studies argue that investing in education can improve parents’ health by relaxing family budget constraints, optimizing the healthy behaviors of elderly people, and strengthening intergenerational non-economic links [32,33,34]. These effects are mainly due to improvements in offspring education, which can significantly increase the effect type, frequency, and extent of financial support that children can eventually provide for their parents [16]. Additionally, it also reduces the labor supply of elderly parents and do not need to continue working in their old age [17]. This has a beneficial effect on their health by reducing wear and tear on their bodies and by freeing up more time for leisure. This also increases the quality of parents’ mental health and life because their children will be able to provide more emotional support during frequent and intensive informal care [12,13]. If offspring are more successful and affluent because they receive more education, they may have more resources to finance and boost the expenditures of the elderly and help parents increase their access to cleaner living environments and household utilities such as a toilet and (piped) running water [26]. As children are the main caregivers for the elderly, an increase in offspring education can also share health knowledge [18] and skills [35] with parents, such as healthy eating habits and improved measures related to disease prevention. Children can improve parents’ healthy behaviors in the long-term care system by providing them with better quality health care services, thereby increasing parents’ health productivity [13]. All of these mechanisms lead to improvements in parents’ health [36].

### 2.2. The Spillover Difference

Although the previous papers analyzed the spillovers of offspring education, they only studied the average effects on the health of the elderly [13,18,26,27,29,36]. Resource substitution theory highlights that educational resources are more important among disadvantaged individuals [37]. Individuals with fewer social or economic resources can use education as an alternative. For example, those with poorer health may feel the benefits of education more than those with better health. The law of diminishing marginal utility thinks that, as one’s health gradually improves, the marginal benefits of spillovers of offspring education gradually decrease. The health of older people has high heterogeneity compared to that of younger people [25]. Changes in health cannot be adequately demonstrated by the average effects alone [38]. A policy is more popular if it has the same average effects on health outcomes as another policy while better improving the health of those who are in poorer health. Therefore, when evaluating the actual effects of policy, we tend to pay more attention to its distributed effects. Most studies have assumed that these spillovers are homogeneous among elderly people with different levels of health. They ignored the possible differences in the spillovers of offspring education on parents with different levels of health, particularly those with disadvantaged health. The deterioration of the health increases daily care needs and medical expenditures of the elderly. This leads to an increase in their reliance on support from offspring at older ages [29]. The positive effects of increased education of offspring are more sensitive in this type of elderly group. Thus, whereas evidence of the positive effects of offspring education on parents’ health is clear, we know less about whether the spillovers change as parents’ health change. This has not received enough attention in previous studies.

### 2.3. Current Study

Our study exploits the 1999 university enrollment expansion policy in China, which decreased the limitations for admission to universities, to estimate the causal relationship between offspring education and parents’ health. Given that offspring characteristics have been shown to be critical for health among older people in China [16,17,30], we controlled for the socioeconomic characteristics of parents and offspring, factors that could influence their health and the educational attainment. We also assessed whether spillovers are heterogeneous across different dimensions of the parents’ frailty index apart from examining relationship. While making up for the lack of existing research, this work also provides a realistic basis for a better analysis of the reduced elderly health inequities caused by increased schooling of their children. We also tested whether these associations differ between mothers and fathers. Next, we analyzed which types of living arrangements can better promote the spillovers of offspring education. This helped us to understand the ways in which education produces positive externalities within the family. Last, we verified the robustness of the research results by employing different measures of parents’ health and offspring education, constructing a placebo test and alternative specification.

### 2.4. Hypothesis

After a review of existing studies combined with an analysis of the problems that remain to be solved, we proposed the following hypotheses:

**Hypothesis** **1** **(H1).**
*The implementation of education policies improves parent*
*s’ health by increasing the years of schooling that their offspring receive. After controlling for the socioeconomic characteristics of the parents and their offspring, this association remains robust.*


**Hypothesis** **2** **(H2).**
*Considering the high heterogeneity of the health of elderly, we believe that the spillovers of offspring education will not be the same among parents with different levels of health. Based on resource substitution theory and the law of diminishing marginal utility, we hypothesize that parents with poorer health are more positively affected. This hypothesis has great potential to reduce health inequality among the elderly.*


**Hypothesis** **3** **(H3).**
*There are differences in the extent of the association between offspring education and the health of fathers and mothers. Given the relationship that mothers have with the labor market and in the family is different from that of fathers, we guessed that mothers are more sensitive to the increase in the years of schooling that their children receive.*


**Hypothesis** **4** **(H4).***The spillovers of offspring education are not the same across different living arrangements. Compared to separation from their children, cohabitation may be more helpful for improving the healthy behaviors of parents and for providing more social support or strengthening emotional communication between generations. Cohabitation also can increase the private transfers of economic resources from offspring to the elderly. These are essential for reducing poverty among the elderly* [39].

## 3. University Enrollment Expansion in China

Before continuing with the research in this article, it was necessary for us to conduct a comprehensive understanding of educational policy reform and institutional circumstances in China. With the development of social economy, China experienced the rapid improvement in education attainment in the late 20th century. Since China resumed college entrance examinations in 1977, the scale of college enrollment has increased year by year. However, by the mid to late 1990s, the admission rate to colleges in China was only around 30%. This rate is not only lower than that of Japan, South Korea, and Singapore, but it is also lower than that of India, which had lower income in the same period. In addition, the Asian financial crisis that broke out in 1997 and the reform of Chinese state-owned enterprises that began in 1998 had a certain degree of impact on China’s economy and employment. In November 1998, Chinese economist Min Tang made the suggestion to the Chinese government that college enrollment be expanded. He argued that the number of undergraduates in China was far lower than that in other countries with a similar level of development. Chinese colleges had the capacity to accommodate more students. He also stated that expanding university enrollment would be able to effectively prevent employment competition between the large number of laid-off workers caused by the reform of state-owned enterprises and young people. It would also be able to stimulate economic growth, expand education consumption, and increase human capital investment in China. Subsequently, the Chinese government issued the “Action Plan for the Promotion of Education in the 21st Century” in 1999, which had the goal of expanding domestic demand, stimulating consumption, promoting economic growth, and relieving employment pressure. This action plan proposed that the college gross enrollment rate should reach 15% by 2010. Since then, efforts to expand enrollment in Chinese colleges have proceeded. This marks the transition of Chinese college education from the elite stage to the mass stage.

Some scholars hold opposing views on the university enrollment expansion policy, believing that it will affect the quality of college education. However, this policy also allows China to achieve a high level of human capital support, which it can use to transform its economic development model and maintain a medium- to high-speed economic growth. The university enrollment expansion has allowed more students to access college education. In 1999, Chinese colleges enrolled 1.6 million students, an increase of 0.52 million or 48% from 1998. College admission rates have since increased from 34% to 55%. The average annual growth rate of college admissions from 1998 to 2019 exceeded 10%. Benefiting from the increase in the number of college admissions created by the university enrollment expansion policy, more than 8 million students have entered college each year since the policy’s introduction (see Figure 1). The university enrollment expansion also has its skeptics. Some of them believe that college education expenditure will squeeze out other family expenditures, thereby increasing the financial burden of the family. In rural and low- and middle-income areas, college education expenditure is an especially heavy financial burden. In order to better maintain family life, parents have to increase their working hours to obtain more income, resulting in a hidden increased danger of health problems in old age. However, little is known about what is the association between the increased education of children as a result of the university enrollment expansion and the health of their parents in old age. These are also questions that we want to research in this paper.

## 4. Data and Variables

### 4.1. Data

We used data from the China Health and Retirement Longitudinal Study (CHARLS) for the analysis. CHARLS uses the Probability Proportionate to Size Sampling method. It covers basic personal information, socioeconomic backgrounds, family structure, health, living arrangements, and other information provided by the respondents and providing important data support for our analysis of the spillovers of offspring education and the quantile treatment effect (QTE).

Most families have more than one child. However, there is no clear evidence determining which offspring has a greater impact on their parents’ health. Therefore, previous studies have used different methods to measure offspring education [17,18,24,26,40]. Using data from the multiwave Survey of Health and Living Status of the Elderly in Taiwan, Zimmer et al. [40] found that offspring with higher educational achievement may have higher socioeconomic status and more health knowledge than siblings with the lowest education or the average educational level of all. Children who receive a higher level of education also have the ability to provide more resources and support to their parents. Therefore, we considered the child with the highest level of educational achievement. This facilitates a more accurate identification of the association between offspring education and parents’ health. If two or more children in a family had the highest educational achievement among their siblings, then we selected the older child for the study. According to the World Health Statistics report released by the WHO, the Chinese life expectancy was close to 80 in 2018. In China, the current minimum age at which someone is eligible to receive their pension is 50. We limited the age of the parents to a range from 50 to 80 years old. To ensure that this paper discusses the highest educational achievement obtained, we deleted the sample of children who are still in school and limited the age to 25 years old and above. We restricted children and parents to this age range to avoid our results being affected by concurrent social and economic changes that may affect the education of children or the health of their parents in the treatment and control groups differently. We included respondent characteristics that are correlates of health outcomes as well as educational attainment in the analysis. We mainly used survey data from 2018 and selected partial information about parents and their children from the 2013 and 2015 data as a supplement and excluded respondents who were missing information on major variables. This means that our sample is very representative of the population. Finally, we obtained 7.118 samples for analysis. In the robustness checks section, we further tested the representativeness of our samples and the sensitivity of our results to different sampling criteria.

### 4.2. Dependent Variables

We used the frailty index to comprehensively reflect the old-age health of the parents. The frailty index describes the health of a given individual by the proportion of unhealthy indicators in all health indicators, which cover multiple aspects, including physical and psychological aspects [41]. As a comprehensive health indicator, the frailty index has no uniform standard for the number of health indicators covered and not sensitive to specific health indicators [42]. Respondents with a lower frailty index have better health. There were six specific indicators that were chosen for the frailty index in our article: self-rated health, happiness index (including respondent’s satisfaction with their life and their own health), number of difficulties in cognitive limitations (including the respondent’s answers to the following questions: What is the year? What season is it? What is the date? What is the day of the week? What is the month? How would you rate your memory at the present time? and successive subtraction calculation (up to five times)), depression (CESD 10) score, number of difficulties in instrumental activities of daily living (IADLs) (including homemaking, preparing a hot meal, shopping for groceries, taking medication, and managing money), and having one or more functional limitation (including difficulties with the following: walking 1 km, getting up from a chair after sitting for a long period, climbing several flights, stooping, kneeling or crouching, extending both arms, lifting or carrying weights, and picking up a small coin from a table). In addition, we also refer to the method employed by Lei et al. [43] to standardize the dispersion of the frailty index. The frailty index used in our article involves a total of 32 health indicators.
(1)$$FI_{i}=\frac{\sum_{\varphi =1}^{n}d_{\varphi i}}{n},$$

In Equation (1), $FI_{i}$ is the frailty index of respondent *i*. $d_{\varphi i}$ is the classification of respondent *i* on the health indicator φ. When φ has a healthy status, $d_{\varphi i}=0$. Otherwise $d_{\varphi i}=1$. The frailty index is a continuous variable with a value range of 0–1. In supplemental analyses, we tested other measures of parents’ health (see the robustness checks section).

### 4.3. Independent Variables

We used the years of completed formal schooling to measure offspring education as an independent variable. The number of years taken to obtain a certain degree is not continuous. There is a certain jump in the number of years of completed formal as the level of the degree increases. Formal schooling is composed of illiteracy (0 years), did not finish elementary school (3 years), home school (3 years), elementary school (6 years), middle school (9 years), high school (12 years), vocational school (12 years), two-/three-year college/associate degree (15 years), and four-year college/bachelor’s degree or above (16 years). We also tested how our results changed when using different measures of offspring education (see the robustness checks section).

### 4.4. Control Variables

Control variables include demographic characteristics variables, socioeconomic variables, and lifestyle variables that are found to be strongly correlated with health in an older age cohort. Demographic characteristics variables include gender (gender), age dummies (age), marital status (marriage), health in childhood (srh15), education (own education), and the number of offspring who are still alive (number of children). We also controlled for the children’s gender dummy (children’s gender), with daughters being assigned a value of 1. Socioeconomic variables included Medicare (Medicare), household income (lnincome), and net current assets (lnasset). Lifestyle variables included type of residence (urban), residence environments (environment), drinking alcohol (drink), and living arrangement (living). Taking the CHARLS survey regarding household living arrangements into account, we divided into two types: cohabitation and separation (Gender: Equal to 0 if the respondent is male and 1 otherwise. Age: Age dummies are set at 5-year intervals. Equal to 1 if the respondent is between 50 and 54 years old and Equal to 2 if the respondent is between 55 and 59 years old, and so on. Marriage: Equal to 1 if the respondent is married and 0 otherwise. Srh15: Equal to 1 if the respondent is healthy at the age of 15 and 0 otherwise. Children’s gender: Equal to 0 if the child of the respondent is male and 1 otherwise. Medicare: Equal to 1 if the respondent has any kind of medical insurance and 0 otherwise. Lnincome: The logarithm of the respondent’s household income. Income includes one or more type of household wage income and individual-based transfers, household agricultural income, self-employed activities income, and household public transfer income. Lnasset: The logarithm of the net current assets of the respondent’s household. Assets include cash, electronic money, deposit in financial institutions, bonds (such as Treasury bills), stocks, and funds, etc. Debt includes loans, credit card balance, etc. Urban: Equal to 1 if the respondent lives in an urban and 0 otherwise. Environment: The score of the respondent’s residence environment. Drink: Equal to 1 if the respondent is a drinker and 0 otherwise. Living: Equal to 1 if the respondent lives with their offspring and 0 otherwise). Due to the differences in the cultural, social, and economic development among different provinces in China, we also controlled for the fixed effect of the province in which the child was born to capture the time-invariant characteristics of province of birth. All the control variables were treated as being exogenous.

## 5. Empirical Strategy

### 5.1. OLS Estimates

For OLS estimates, we assumed that the parents’ health is a function of offspring education and other variables.
(2)$$FI_{i}=\beta_{0}+\beta_{1}Edu_{i}+\beta_{2}X_{i}+\lambda_{m}+u_{i},$$

In Equation (2), $FI_{i}$ is the parents’ frailty index. $Edu_{i}$ is the education achievement attained of the highest-educated child, which is measured in years of schooling. $X_{i}$ is other control variables. It includes the demographic characteristics variables, socioeconomic variables, and lifestyle variables of the elderly and their offspring. $\lambda_{m}$ is the fixed effect of the child’s birth province, and $u_{i}$ is a random error term. $\beta_{0}-\beta_{2}$ are the coefficients to be estimated, where $\beta_{1}$ measures the marginal effect of each additional year of offspring schooling on the parents’ health.

However, the OLS estimates are biased due to the endogeneity of education. In the OLS estimates, factors that affect education and health at the same time, such as unobservable innate ability, family background, etc., are included in $u_{i}$. This results in omitted variable bias. For example, offspring with higher learning endowments may acquire a wider range of health knowledge and skills while receiving higher educational achievement, thus improving their parents’ healthy behaviors. Meanwhile, the OLS estimates may lead to a reverse causality problem, where healthier parents have more resources to devote to their health and their offspring education. Omitted variable bias and reverse causality lead to the endogeneity problems, resulting in bias errors in the OLS estimates of $\beta_{1}$. To improve the causal interpretation of our study, we employed an instrumental variable (IV) regression/two-stage least square (2SLS) approach to further examine the relationship between the two parties. To explain the correlation between offspring education and parents’ health within the same province, we clustered our standard errors at the provincial level [17]. Meanwhile, we also tested the influence of different standard errors, including clustering standard errors at the community and city level and the robust standard errors approach on the results (the different methods provide a similar size of standard error, so these did not change the results qualitatively).

### 5.2. IV Estimates

In order to eliminate the problem of endogeneity, we used university enrollment expansion in China as the instrumental variable of offspring education and employed the IV/2SLS approach to estimate the association between offspring education and parents’ health. These policy changes implied that in 1999, children aged 18 and under would have more opportunities to enter universities and realize their college dreams, whereas children older than 18 years of age would not be required to do so. Researchers often use quasi-natural experiments, for example educational policy reform, to solve the endogeneity of education. Most of the previous relevant studies chose compulsory education reform as the exogenous instrumental variable [17,18,26,29]. For instance, based on the compulsory education reform in Tanzania in 1974, De Neve and Fink [26] found that the reform not only greatly improved the educational achievement of the beneficiary groups, but also improved the survival probability of their parents, especially when the welfare sectors were underdeveloped and within-family transfers were common. Lundborg and Majlesi [18] used the Swedish compulsory education reform rolled out in 1948 and found that the years of offspring education had no significant effect on parents’ longevity. However, this hides substantial heterogeneity caused by gender. Their further analysis found that the level of education achieved by daughters affected the longevity of their fathers, especially those with lower socioeconomic backgrounds. Using both temporal and geographical variation during China’s compulsory education reform policy implementation from 1985 to 1991, Ma [17] found that schooling improved various health and cognitive outcomes for older parents. These were mainly achieved by supplying financial support, influencing access to resources, affecting the labor supply and psychological well-being of parents. In addition, Ma et al. [29] also used a similar method to analyze the impact of Mexico’s compulsory education reform. Offspring education improved parents’ cognitive abilities in verbal learning, verbal fluency, and orientation. Mothers may benefit more from the schooling of adult children than fathers. These studies provide a beneficial reference for our analysis.

However, as China’s overall education continues to rise, the impact of university enrollment expansion policy gradually becomes more evident [44] compared to compulsory education law. Figure 2 shows the average years of schooling that children received in the 2018 CHARLS survey. Scatter diagram and quadratic fitted lines are drawn separately for the amount of schooling that children born before 1980 and those born in 1980 or later. Consistent with previous research [45], Figure 2 implies a jump in education for the 1980 cohort among offspring. This supports our empirical strategy of using the changes induced by the university enrollment expansion in China as being an instrumental variable to resolve the endogeneity of education. It has improved the education of the cohorts affected by the policy, but it has not affected the education of other cohorts. In addition, exposure to this policy not correlated with the determinants of parents’ health outcomes and the unobserved omitted variables. Therefore, we used 1980 as the cutoff birth cohort to identify offspring who had been exposed to the policy. Specifically, the model is set as follows:(3)$$Edu_{i}=\alpha_{0}+\alpha_{1}Ref_{i}+\alpha_{2}X_{i}+\lambda_{m}+u_{i},$$
(4)$$FI_{i}=\beta_{0}+\beta_{1}Edu_{i}+\beta_{2}X_{i}+\lambda_{m}+u_{i},$$

$Ref_{i}$ indicates whether children were affected by the university enrollment expansion policy in 1999, which was used as the instrument for years of schooling. The first stage is given by Equation (3), where $Ref_{i}$ equals 1 if the highest-educated offspring was born in or after September 1980 and was affected by the policy reform. We define this cohort as the treatment group. $Ref_{i}$ equals 0 if the child was born before September 1980. We define this cohort as the control group [45]. The university enrollment expansion had different effects on the children belonging to different age cohorts in the same province. If there are significant differences in education among children of different age cohorts in the same province, this outcome can be considered to be the impact brought about by the university enrollment expansion policy after excluding other influencing factors. Therefore, we constructed $Ref_{i}$ to accurately measure the impact of the university enrollment expansion on offspring education.

### 5.3. IVQR Estimates

Since Koenker and Bassett [46] proposed the statistical characteristics and estimation methods of conditional quantiles, this method has been widely used in studies because of its ability to capture the effects of heterogeneity. The characteristics of conditional distribution are described comprehensively by means of the parameter estimates for different quantiles. Compared to linear regression, it relaxes the assumption of the distribution of random perturbations. For models with heteroscedasticity, the estimator is not susceptible to outliers. This means that the regression model has strong robustness. The classic quantile regression model assumes that the key variables are exogenous. However, the variables that are studied in reality are often endogenous. Therefore, Chernozhukov and Hansen [47,48] further proposed a non-parametric instrumental variable quantile regression model to describe QTE. IVQR estimates are an important and powerful tool for characterizing distribution effects. Compared to OLS estimates and 2SLS estimates, the two biggest advantages of IVQR are that it is not only able to describe the distribution effects of the dependent variable, but it can also estimate the parameters by minimizing the absolute offset and overcoming the influence of outliers.

Suppose we use a binary variable Z that can be used as an instrumental variable. We can then define two classes of potential treatment variables D. D_1_ is the treatment effect of the individuals affected by the policy. D_0_ is the treatment effect if it is not affected by the policy. The causal effects that we are interested in are defined by the counterfactual described by potential outcomes and potential treatment effect. Potential outcomes include Y^1^ and Y^0^, representing the individual’s outcome variable value at D_1_ and D_0_, respectively. The goal of causal inference is to understand the distribution characteristics of Y^1^ and Y^0^ over the sample. The quantile treatment effect $\theta^{\tau }$ can be uniformly estimated by weighted quantile regression as follows.
(5)$$\left(\hat{\beta_{IV}^{\tau }},\hat{\theta_{IV}^{\tau }}\right)=argmin_{\beta ,\theta }\sum W_{i}^{AAI}\times \rho_{\tau }\left(Y_{i}-X_{i}\beta -D_{i}\theta \right),$$

$W_{i}^{AAI}$ is the weight index, and its calculation formula is:(6)$$W_{i}^{AAI}=1-\frac{D_{i}\left(1-Z_{i}\right)}{1-P\left(Z=1|X_{i}\right)}-\frac{\left(1-D_{i}\right)Z_{i}}{P\left(Z=1|X_{i}\right)},$$

According to the observed *Z* and *X_i_*, we can first calculate $P\left(Z=1|X_{i}\right)$. Then, we can calculate $W_{i}^{AAI}$ by substituting $P\left(Z=1|X_{i}\right)$ into Equation (6), and estimate $\theta^{\tau }$ by substituting $W_{i}^{AAI}$ into Equation (5).

In addition, we used the bootstrap self-help method to sample and estimate the standard deviation of the parameter estimator. We repeated the sampling 100 times. We also used the method proposed by Mourifié and Wan [49] to test the validity of the instrumental variables (not shown here), and found that it is effective to use college expansion as the instrumental variable of offspring education.

## 6. Results

### 6.1. Descriptive Statistics

Table 1 shows the descriptive statistics of the major variables. It can be seen that the elderly parents’ frailty index is low, with an average value of 0.365. This indicates that the elderly population is relatively healthy overall. The average respondents in the sample scored 0.762 for self-rated health, 0.387 for the happiness index, and 4.622 for the CESD 10. The average IADLs score of the respondents was 0.453. The average number of difficulties in cognitive limitations of 3.008 and of one or more functional limitations of 2.093 in the sample. The proportion of female in the sample was 52.4%. Almost 27.8% of them lived with their offspring and 38.7% of them lived in urban areas. Each family had an average of three living children. The average age of the elderly parents in the sample was 64. The children with the highest level of education had spent an average of nearly 11 years in school, compared to the 5.469 years of schooling that their parents received. It can thus be seen that although educational reform of China has made remarkable achievements, the average number of years of education per capita still needs to be improved. The development of education in China still faces many severe challenges and problems that need to be solved urgently.

### 6.2. OLS Estimation Result

First, we estimated the coefficients from Equation (2) based on the OLS approach. Table 2 reports the OLS estimation results of the spillovers of offspring education on the parents’ frailty index. It can be found from Table 2 that offspring education was negatively and significantly associated with the parents’ frailty index. However, the effect size was small: One additional year of offspring education decreased the frailty index of the elderly parents by 0.006 or about 1.64%. The estimated results of the control variables essentially conform to theoretical expectations. Not surprisingly, well-educated parents also tend to be healthier. An additional year of parental education was associated with a 0.025 decrease in the frailty index after accounted for offspring education. The health of males was better than that of females. There was also a positive correlation between the parents’ health before 15 years of age and the in old age. In addition, we found that the increase in the number of children was a disadvantage to the health of the elderly. The results of the subsample showed that the spillovers of offspring education are almost the same in the cohabitation and separation cohorts.

### 6.3. IV Estimation Result

Table 3 shows results of the first stage regression of the 2SLS estimates. It can be seen that the coefficient of the instrumental variable is positive and significant at the 1% level in both the whole sample and in the sub-sample of respondents with different living arrangements. The results indicate a positive and significant effect of exposure to the university enrollment expansion policy on the years of children’s schooling. The implementation of this policy increased the average number of years of schooling for children by 1.342 years in the treatment group. For the sub-samples of children in terms of cohabitation versus separation, the university enrollment expansion increased the average years of schooling by 0.870 and 1.509 years, respectively. Based on the Kleibergen–Paap rk statistic, the F statistics of weak identification test from the first stage are well above 10, which indicates the strength of the instrument [50]. In addition, the results of the Hausman test show that there is indeed an endogeneity of offspring education (*p* < 0.01). We therefore took the IV estimation result as the preferred specification and the OLS estimation result as reference.

Table 4 reports the results of the second stage regression of the 2SLS estimates. Compared to the OLS estimation results, the absolute value of the coefficient of offspring education in the 2SLS estimation results is great. This suggests that the omitted variable and reversed causality problems biased the OLS estimation. The estimation results of the whole sample show that each additional year of offspring education reduces the frailty index of the elderly by about 0.017 or about 4.66% of the baseline average, compared to a 0.006 decrease in the equivalent OLS estimation results. The sub-sample results also prove that the parents’ frailty index is lower when their children are in school for longer. These spillovers are higher when parents were found to be cohabiting with their children instead of separating. We believe that cohabitation may allow the elderly to receive more financial support, life care, and spiritual counsel from their children in a developing country such as China compared to parents who live apart from their children. This also verifies the viewpoint of the intergenerational support theory [51]. The estimated results for the control variables are similar to the OLS estimates with the exception of parents’ own level of education. The dimensions of the coefficients for the parents’ own education are reduced in IV models and are not significant compared to the OLS models. However, the two coefficients are not comparable given that estimates of the parents’ own education are biased due to potential endogeneity. The 2SLS estimation results indicate that the correlation between the health of the elderly and offspring educational attainment is stronger than that with their own education. Table A1 also shows the spillovers of offspring education on the various health variables that make up the frailty index, which is important when designing public policies that are aimed at the specific health problems of the elderly.

However, these findings are divided between fathers and mothers. To assess whether mothers or fathers benefit differently from the education of their children, we estimated the spillovers using the IV models for males and females separately. The spillovers on the health of the mother are almost two times that on the father (0.022 vs. 0.011), which is equivalent to a decrease of 5.34% and 3.51%, respectively, as shown in Table 5. The relationship between the two is basically similarity in different living arrangements (9.83% vs. 7.74%, 4.54% vs. 2.54%). These may reflect conjecture from the relationship between the labor market and the family unit. Females are at an employment disadvantage compared to males in the labor market. It is no secret the females have lower labor participation rates in most industries than male. This causes females to rely more on the family resources as well as those of their children. Besides this, fathers are more likely to play a strict role in family education, while mothers are more likely to play an approachable role. Furthermore, in the terms of raising children and even grandchildren, mothers assume the main care responsibilities [52,53]. Therefore, the relationship between children and their mothers is different from that with their fathers, maybe closer, which makes the impact of offspring education on the elderly health of fathers and mothers different. This has not received much attention in previous studies. All in all, both the OLS estimates and 2SLS estimates prove that offspring education can significantly improve parents’ health, especially that of mothers. Due to differences in physical conditions and social status, elderly females are exposed to a greater risk of health decline although the average life expectancy is relatively high for elderly females [54,55]. Therefore, this result has deeper significance; that is, the education of children can not only improve the health of the elderly, but it can also reduce the gender inequalities in health.

### 6.4. IVQR Estimation Result

It can be seen from Panel A of Figure 3 that the quantile treatment effect of offspring education on the parents’ frailty index is negative in both the whole sample and in the sub-sample. This is consistent with the results of the OLS estimates and 2SLS estimates. However, these effects are not the same in the different quantiles of the frailty index. For the elderly, the decrease in the frailty index means the improvement in health to some extent. The spillovers of offspring education on the parents’ frailty index fluctuates significantly with the decrease in the quantiles. There is an overall decreasing trend. At the ninetieth quantile, the spillovers are 0.029. As the quantile decreases, the spillovers gradually decrease and are 0.017 at the median point (i.e., the fiftieth quantile). At the tenth quantile, the spillovers further decrease to 0.003. As the decrease in the parents’ frailty index (i.e., the improvement of health), the spillovers gradually decrease. The spillovers affecting the parents from the highest quantile are nearly 10 times those affecting the lowest quantile. This means that the spillovers of offspring education are greater for the elderly with poorer health. When one’s children attain higher levels of education, it contributes to narrowing the health inequality between elderly with poorer health and with better health.

Figure 3 also shows the quantile treatment effect of the spillovers on different living arrangements. The results are similar to the whole sample. Even under different living arrangements, the spillovers of offspring education have shown declining characteristics as the parents’ health increases. This means that the spillovers have strong robustness. This once again confirms the importance of offspring education in reducing health inequality of the elderly. From Panel A of Figure 3, it can also be seen that the spillovers when two generations cohabiting are larger than when they separating. Under the different living arrangements, the spillovers of offspring education are 0.044 and 0.024, respectively, in the ninetieth quantile. As the quantiles decreases, the spillovers gradually decreased to 0.030 and 0.016, respectively, at the median point (i.e., the fiftieth quantile). At the tenth quantile, the spillovers further decreased to 0.005 and 0.003, respectively. As the quantiles decline, the gap between cohabitation and separation gradually decrease and tend to be the same at the lowest quantile. To sum up, the spillovers of offspring education on the health of parents with different frailty indexes are non-linear and non-averaged. This effect is also different according to different living arrangements. Offspring education has improved the health of disadvantaged cohorts more and is likely to be one of the important mechanisms for promoting health equality. Panels B and C in Figure 3 present similar results of mothers versus fathers in terms of the frailty index. The spillovers of offspring education decrease as the health of mothers and fathers improves. Although fathers also benefitted from having more-educated children with respect to their health outcomes, the frailty indexes of the mothers were more sensitive to the offspring educational attainment. Furthermore, we also found that parents can benefit from more health outcomes resulting from the spillovers of offspring education when two generations cohabiting together.

## 7. Robustness Checks

### 7.1. Measure of Parents’ Health and Offspring Education

The OLS and 2SLS estimation results both show that offspring education has significant spillovers on parents’ health, and these effects are not the same for fathers and mothers. If this conclusion is robust, then we can use different indicators to measure parents’ health and offspring education that will not have a major influence on the overall estimation results. Subjective life expectancy reflects the conditional expectations of specific individuals under a given set of information, including health status and behavior habits [56]. It can more accurately predict mortality [57]. To further verify the robustness of the above conclusions, we use subjective life expectancy as a measure of parents’ health. The estimation results shown in columns 2 and 3 of Table 6 demonstrate that increased offspring education can significantly improve the subjective life expectancy of their parents. However, the spillovers on the subjective life expectancy of fathers are greater than on mothers, which is the opposite of the OLS and 2SLS estimation results. The estimation results did not significant changes. The coefficients of the other control variables are basically the same as those noted above.

In the previous analysis, we retained the children with the highest level of educational achievement. Since most families have more than one child, the findings on well-educated children may not be applicable to other children. Therefore, we referenced the methods of Lundborg and Majlesi [18] and Ma et al. [29] to analyze whether the above results can be applied to children with the least-educated and the average educational achievement among all living siblings. Alternative measures of offspring education render similar associations with parental health in columns 4–7 of Table 6. However, the dimension of the spillovers varied in both groups compared to in the results for children with the highest educational achievement. This also confirms previous studies, such as the one conducted by Zimmer et al. [40], which showed that children with different levels of educational attainment may be different in terms of influencing their parents’ health. Children with the highest level of educational attainment, and therefore with more resources, have the greatest impact on the health of their parents [40]. Therefore, parents in China may rely more on the resources provided by the child with the highest level of educational achievement in their old age rather than all of their offspring.

### 7.2. Placebo Test

We also constructed a placebo test based on the year in which the counterfactual policy took effect. First, we assume that the university enrollment expansion policy was implemented two years after 1999 (i.e., 2001). We believe that if the estimation results of our study are the impact of university enrollment expansion rather than the time trend, then we should not come to the same conclusion regarding the hypothetical years of university enrollment expansion. Therefore, we used 16–17-year-olds as the assumed treatment group (real ages are 14–15 years old) and the 18–19-year-olds as the assumed control group (real ages are 16–17 years old) to estimate in the virtual effective year of the policy. The second column of Table 7 provides the estimated value of the first stage and the test of the instrumental variable. The first line shows that there is no significant difference in terms of the years of schooling between the assumed treatment group and the assumed control group.

Similarly, we also assumed that the implementation of the university enrollment expansion policy occurred two years before 1999 (i.e., 1997). Then, we used the 16–17 years old as the assumed treatment group (real ages are 18–19 years old) and the 18–19 years old as the assumed control group (real ages are 20–21 years old) to estimate in the virtual effective year of the policy. The estimation results, which are shown in the third column of Table 7, once again indicate that no significant differences can be observed in terms of the years of schooling between the assumed treatment group and the assumed control group.

The last line of Table 7 provides the estimation results of the second stage of 2SLS, showing that offspring education has no significant relationship with the parents’ health in the placebo test. Moreover, using the placebo test indicators render the F statistics of the weak-identification test lower than the rule of thumb value of 10, indicating that there is a weak instrument problem. The *p*-values of Anderson–Rubin Wald test indicate that the coefficients for the offspring education are not statistically different from zero in the placebo test. Therefore, the placebo test results show that the estimation results in our study are not caused by the time trend. This reconfirms the validity of our conclusion.

### 7.3. Specification and Bandwidths

We also tested the sensitivity of our results to alternative specifications for different ranges of child’s and parents’ ages to ensure that exposure to the college expansion satisfies the exclusion restriction. Given the wide age range of child and hence the wide age range of parents, the college expansion may have highly correlated with all kinds of other cohort trends and changes. The validity of the instrument could be compromised, and the IV estimates could be inconsistent. Therefore, we restricted the sample of the two bandwidths relative to the 1980 cohort: [−15, 15] and [−10, 10]. The result of shortening the bandwidth of the children is basically to the same as that of the OLS and 2SLS estimations. Moreover, the reason for limiting the age to 80 and below in the previous analysis was that the average life expectancy for the Chinese population in 2018 was close to 80 years old. Given that the healthy life expectancy of the Chinese population in 2018 was 68.5 years, we narrowed the parental age range to 50 to 70 years. Healthy life expectancy is derived from life expectancy, which is based on population mortality and the prevalence or health status data to estimate the life expectancy of healthy for a certain age group. Healthy life expectancy can reflect both longevity and the degree of health. The WHO defines it as the life expectancy of an individual in good health, which is equivalent to the average expected number of years that a person is able to live in a healthy state [58]. The results in Table 8 and Figure 4 show that the magnitude and tendency of the spillovers are similar to those of the main results.

## 8. Discussion

Many studies have assumed that the intergenerational mobility of human capital, including health and education, is usually a one-way spillover of older generations to younger ones [25]. However, it is possible that the link runs in the opposite direction as well [18]. Although the positive impact of education on human health has been well established in the literature, the causal effect of offspring education on the health of the older population has only recently received attention [12,16,17,32,40]. Our study expanded on recent work assessing the relationship between offspring education and parents’ health inequality by providing causal evidence in China [17] in a number of important ways. The present research takes advantage of the exogenous policy changes caused by the 1999 college expansion in China to construct instruments for offspring education and compares results with findings using standard OLS models. Using policy reforms as an IV helps to reduce the endogeneity bias that is frequently present in the growing body of work examining the spillovers of offspring education [17,18,26,29]. The first-stage results in the IV estimation show that the university enrollment expansion policy has significantly increased the years of schooling that young people in China receive, with about 1.342 years. This proves the importance of the university enrollment expansion for the development of human capital in China [44,45,59]. Furthermore, the second-stage results from the IV estimation show that the education of children has significant upward spillovers on parents’ health. When we control for the socioeconomic characteristics of the elderly and children, this association is robust. Specifically, each additional year of offspring education generates about a 4.66% or 0.017 decrease in the frailty index of parents. OLS estimates do underestimate the spillovers of offspring education. A similar situation occurs in other studies [17,26,29]. Because IV regression estimates the local average treatment effect, it pays more attention to those who are the most severely affected by the instruments. In other words, the children most directly affected by the changes in the university enrollment expansion may also benefit more from educational resources in the family [29].

Our results on gender differences suggest that the spillovers of offspring education are more conducive to improving the health of mothers (0.022 vs. 0.011). This discrepancy may be attributed to differences in the relationship between offspring and both parents over the course of family life and the position of parents in the labor market. As the primary caregiver in the family, mothers are less dependent on the labor market. They are more likely formulate closer relationships with children [16,52,53]. This may be the main reason why offspring education is more positive and significant for the health of mothers. This means that offspring education can also promote gender equality in health.

We also tested whether intergeneration support, measured by different living arrangements, helps explain the link between offspring education and parents’ health inequality. We found that the spillovers have a greater influence on parents who live with their children than separate. Offspring living with their parents can keep in frequent contact and help their parents access better quality health care services [13]. This contributes to emotional support between generations [12]. When well-educated children share health knowledge and skills within the family, it can also help parents develop good health behaviors and habits [18]. Resource transfer is also an important way that children can affect the health of their parents. Children can provide more opportunities to support their parents with financial resources in cohabitation [14,32].

In addition, further analysis of the elderly with different frailty index demonstrated that the improvement of offspring education in relation to their health is non-linear and non-averaged. The spillovers on parents of poorer health are greater than with better health. Offspring education suppresses the “Matthew Effect”, which can lead to the further widening of health inequality among different cohorts of elderly people. The spillovers conform to the law of diminishing marginal utility between generations. It is likely to be one of the important mechanisms for promoting health equality. The findings for different living arrangements are similar to the results of the whole sample. However, the spillovers in cohabitation situations are greater than they are when children are living separately from their parents. We analyzed the quantile treatment effect of the intergenerational mobility of human capital between two generations. These results extend the previous findings on health and education [12,16,17,18,29,30,32] and also lend support to resource substitution theory, which suggests that parents with health disadvantages are more sensitive to the resources of their offspring [37].

We recognize that the present study has several limitations that provide important directions for future research. First, our results are cross-sectional, based on the latest CHARLS data from 2018. Therefore, our judgment on the improvements of parents’ health by offspring education is limited to the current period. Future work should use longitudinal data to assess the spillovers of offspring education, especially over a longer period of time. This will be very helpful for analyzing dynamic changes in parents’ health. The longitudinal analysis would help strengthen the argument regarding the spillovers of offspring resources on parents’ health [30,33,36,60].

Second, the majority of the research respondents came from families with multiple children (although we only analyzed one of their children). After the implementation of the one-child policy in China, parents were only allowed to have one child. Although this policy has been gradually abolished in China, more and more young parents still preferred to have only one child as the society development. This allows parents to invest more in education resources for their child [61]. At the same time, the parents can devote less energy to raising their child. This may lead to changes in the association between offspring education and parents’ health. The only child family cohort should be the focus of future research.

Finally, we did not directly measure the underlying mechanisms by which offspring education improved parents’ health, only a simple analysis. Thus, whereas plausible, our analysis on underlying mechanisms is only speculative and should be interpreted cautiously. There is growing interest in how educational resources have spillovers among family members, whose lives are interconnected and interdependent. Existing studies suggest that investment in offspring education can improve the health of the elderly population by relaxing family budget constraints, optimizing the healthy behavior of parents, and strengthening intergenerational non-economic links [16,17,32,33,34]. However, are different mechanisms equally important in explaining the relationship between the two? We do not agree. The pathways linking offspring education to the parents’ health depend on the changing institutional circumstances of Chinese society. In the new institutional circumstances, there may be new potential mechanisms. That is what we’re going to research next.

## 9. Conclusions

Despite these limitations, we provided new causal evidence on the intergenerational mobility of health and education. We believe that education is an important way to increase human capital and that it also plays an important role in improving the health of aging parents as well as the overall health of the family. This study provides a supplemental understanding that offspring education has the dual significance of reducing population and gender inequalities in health. Therefore, we suggest that the government increase its investment in education and optimize the allocation of education resources, especially in rural and low- and middle-income areas where pension resources are relatively limited. Households in these areas have less disposable personal income. The funds that can be used to invest in offspring education are scarce. Increasing the inclination of public education in these areas can effectively reduce the financial burden of families. Meanwhile, it also can exploit the externality of education to improve the health of the elderly and narrow the population and gender inequalities in health. This suggestion appears to support the assumption that the role of offspring education in terms of areas where there is inadequate public support and social security may be more important because a greater proportion of elderly parents depend on their children for care and support [16,25,26,27,32].

Second, colleges should continue to expand the scale at which they enroll students. Over the past 20 years, university enrollment expansion has mainly increased the number of undergraduate graduates [62,63]. Considering the positive impact of college expansion on the health of the elderly, it is possible to selectively increase the enrollment of postgraduate and junior college students. These measures can meet the diversified needs of the elderly in health care. If the expansion of the well-educated population is stalled, this will prevent parents from gaining potentially greater health advantages from the educational attainment achieved by their offspring. Further government investment in education may have a profound impact on reducing health inequality among the elderly.

Third, it is recommended that children live in households with their parents in order to better exert the upward spillovers of offspring education on parents’ health. Our research found that when older parents choose to live with their offspring, the spillovers are more than two times as large as when they live separately. This provides valuable suggestions for the elderly parents, especially when they are in poor health. On the one hand, parents who live with their offspring may receive physical, mental, and financial support from their offspring, which may affect their health. On the other hand, offspring may also receive support and help from their parents. Parents may help take care of their grandchildren or provide daily life care [64,65,66]. These results would seem to suggest that intergenerational communication with offspring may promote health equality in the elderly population. Compared to previous analyses, offspring education may have a far-reaching impact on parents’ health inequality, especially in an aging developing country such as China. Although education policy is unlikely to be the focus of improving the health of the elderly, it might be used as a supplement to public health policy [67] to help reduce social inequalities in health and promote the sustainable development of the public health sector.

## Figures and Tables

**Figure 1 ijerph-19-02006-f001:**
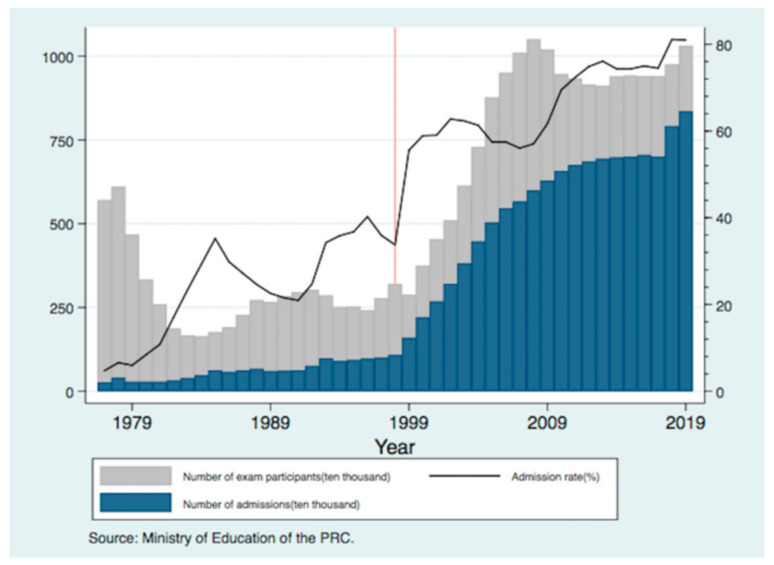
The development of college education in China from 1977 to 2019.

**Figure 2 ijerph-19-02006-f002:**
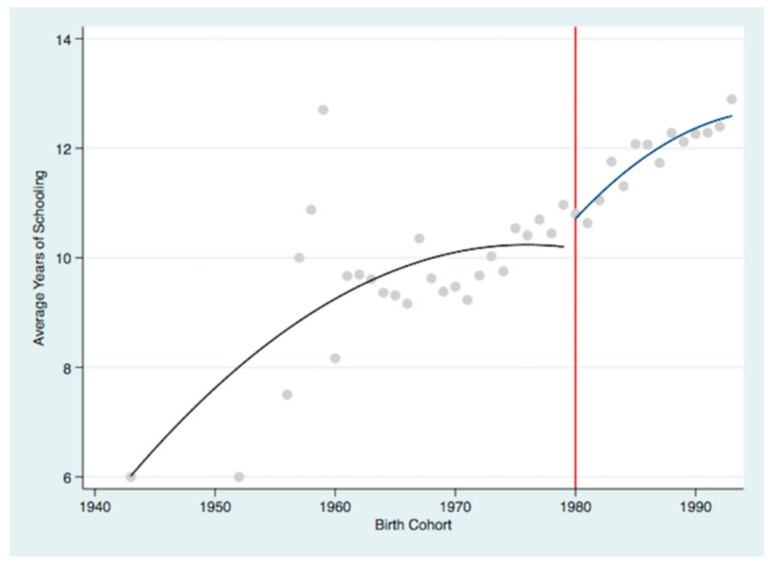
The jump in offspring education before and after the university enrollment expansion.

**Figure 3 ijerph-19-02006-f003:**
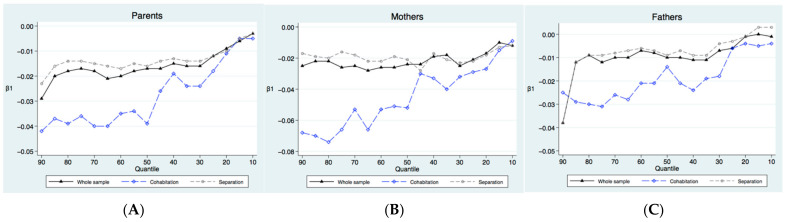
IVQR estimation results of offspring education on parents’ health: (**A**) parents; (**B**) mothers; (**C**) fathers.

**Figure 4 ijerph-19-02006-f004:**
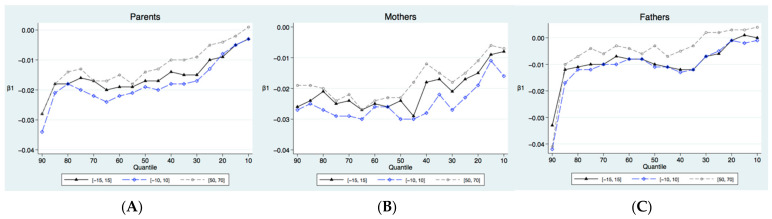
IVQR estimation results of robustness to specification and bandwidths: (**A**) parents; (**B**) mothers; (**C**) fathers.

**Table 1 ijerph-19-02006-t001:** Descriptive statistics.

Variables	Observations	Mean	Standard Deviation
FI	7118	0.365	0.187
Self-rated health	7118	0.762	0.426
Functional limitations	7118	2.093	1.892
IADLs	7118	0.453	1.004
Cognitive limitations	7118	3.008	1.627
CESD 10	7118	4.622	2.922
Happiness index	7118	0.387	0.632
Children’s education	7118	10.935	3.707
Gender	7118	0.524	0.499
Age	7118	64.069	8.006
Marriage	7118	0.980	0.139
Urban	7118	0.387	0.487
Srh15	7118	0.716	0.451
Own education	7118	5.469	4.079
Medicare	7118	0.971	0.167
Lnincome	7118	9.210	1.991
Lnasset	7118	8.775	1.573
Environment	7118	0.577	0.213
Drink	7118	0.365	0.481
Children’s gender	7118	0.428	0.495
Children’s number	7118	2.953	1.541
Living	7118	0.278	0.448

Notes: All data come from China Health and Retirement Longitudinal Study (2013, 2015, 2018).

**Table 2 ijerph-19-02006-t002:** OLS estimation of offspring education on parents’ health.

Variables	Whole Sample	Cohabitation	Separation
Children’s education	−0.006 ***	−0.006 ***	−0.006 ***
	(0.000)	(0.001)	(0.001)
Gender	0.089 ***	0.077 ***	0.095 ***
	(0.006)	(0.008)	(0.007)
Age	0.001 ***	0.002 ***	0.001 **
	(0.000)	(0.000)	(0.000)
Marriage	−0.012	0.016	−0.023
	(0.015)	(0.031)	(0.018)
Urban	−0.020 ***	−0.035 ***	−0.015 **
	(0.006)	(0.012)	(0.006)
Srh15	−0.032 ***	−0.023 ***	−0.035 ***
	(0.003)	(0.007)	(0.004)
Own education	−0.025 ***	−0.031 **	−0.023 *
	(0.008)	(0.014)	(0.013)
Medicare	−0.024 *	−0.049 *	−0.011
	(0.013)	(0.028)	(0.014)
Lnincome	−0.007 ***	−0.003	−0.009 ***
	(0.001)	(0.002)	(0.001)
Lnasset	−0.016 ***	−0.016 ***	−0.016 ***
	(0.001)	(0.003)	(0.001)
Environment	−0.102 ***	−0.075 ***	−0.111 ***
	(0.011)	(0.018)	(0.014)
Drink	0.002	−0.006	0.006
	(0.006)	(0.009)	(0.008)
Children’s gender	−0.013 ***	−0.005	−0.015 **
	(0.004)	(0.009)	(0.006)
Children’s number	0.013 ***	0.015 ***	0.013 ***
	(0.002)	(0.004)	(0.002)
Living	-0.007 *		
	(0.004)		
Constant	0.687 ***	0.627 ***	0.700 ***
	(0.023)	(0.049)	(0.025)
Province FE	Yes	Yes	Yes
Observations	7118	1978	5140
Adj-R2	0.259	0.270	0.255

Notes: Standard errors clustered at the province level. *** *p* < 0.01, ** *p* < 0.05, * *p* < 0.1.

**Table 3 ijerph-19-02006-t003:** The first stage estimation of 2SLS of offspring education on parents’ health.

Variables	Whole Sample	Cohabitation	Separation
Ref	1.342 ***	0.870 ***	1.509 ***
	(0.099)	(0.154)	(0.115)
Control variables	Yes	Yes	Yes
Province FE	Yes	Yes	Yes
Observations	7118	1978	5140
F statistics of weak identification	206.882	31.795	199.148

Notes: The control variables are shown in Table 2. Standard errors clustered at the province level. *** *p* < 0.01.

**Table 4 ijerph-19-02006-t004:** The second stage estimation of 2SLS of offspring education on parents’ health.

Variables	Whole Sample	Cohabitation	Separation
Children’s education	−0.017 ***	−0.032 ***	−0.014 ***
	(0.003)	(0.010)	(0.003)
Gender	0.089 ***	0.081 ***	0.094 ***
	(0.006)	(0.010)	(0.007)
Age	0.001 ***	0.001 ***	0.000 **
	(0.000)	(0.000)	(0.000)
Marriage	−0.004	0.048	−0.019
	(0.015)	(0.033)	(0.018)
Urban	−0.009	−0.007	−0.006
	(0.005)	(0.012)	(0.006)
Srh15	−0.030 ***	−0.026 ***	−0.033 ***
	(0.003)	(0.010)	(0.003)
Own education	−0.008	0.006	−0.011
	(0.009)	(0.022)	(0.013)
Medicare	−0.009	−0.009	−0.001
	(0.012)	(0.032)	(0.014)
Lnincome	−0.006 ***	0.001	−0.008 ***
	(0.001)	(0.003)	(0.001)
Lnasset	−0.013 ***	−0.008**	−0.014 ***
	(0.001)	(0.004)	(0.002)
Environment	−0.060 ***	0.032	−0.081 ***
	(0.015)	(0.049)	(0.018)
Drink	0.002	−0.000	0.006
	(0.007)	(0.011)	(0.008)
Children’s gender	−0.015 ***	0.005	−0.018 ***
	(0.004)	(0.010)	(0.006)
Children’s number	0.011 ***	0.010 **	0.011 ***
	(0.002)	(0.004)	(0.002)
Living	−0.021 ***		
	(0.005)		
Constant	0.718 ***	0.645 ***	0.727 ***
	(0.023)	(0.049)	(0.027)
Province FE	Yes	Yes	Yes
Observations	7118	1978	5140
Adj-R2	0.220	0.093	0.233
*p*-value of Kleibergen–Paap rk LM statistic	0.000	0.001	0.000
Kleibergen–Paap rk Wald F statistic of weak identification test	206.882	31.795	199.148
*p*-value of Anderson–Rubin Wald test	0.000	0.000	0.000
*p*-value of Hansen J statistic	0.000	0.000	0.000
*p*-value of Endogeneity test	0.003	0.018	0.033

Notes: Standard errors clustered at the province level. *** *p* < 0.01, ** *p* < 0.05.

**Table 5 ijerph-19-02006-t005:** Gender differences in spillovers of offspring education on the health of mothers and fathers.

Variables	Mothers			Fathers		
	Whole Sample	Cohabitation	Separation	Whole Sample	Cohabitation	Separation
Children’s education	−0.022 ***	−0.039 ***	−0.019 ***	−0.011 ***	−0.024 **	−0.008 *
	(0.004)	(0.014)	(0.005)	(0.004)	(0.012)	(0.004)
Control variables	Yes	Yes	Yes	Yes	Yes	Yes
Province FE	Yes	Yes	Yes	Yes	Yes	Yes
Observations	3727	1076	2651	3391	902	2489
*p*-value of Kleibergen–Paap rk LM statistic	0.000	0.009	0.000	0.000	0.002	0.000
Kleibergen–Paap rk Wald F statistic of weak identification test	97.517	11.562	105.150	124.073	17.821	98.103
*p*-value of Anderson–Rubin Wald test	0.000	0.021	0.000	0.012	0.003	0.120
*p*-value of Hansen J statistic	0.000	0.000	0.000	0.000	0.000	0.000
*p*-value of Endogeneity test	0.004	0.061	0.023	0.269	0.075	0.845

Notes: The control variables are shown in Table 2. Standard errors clustered at the province level. *** *p* < 0.01, ** *p* < 0.05, * *p* < 0.1.

**Table 6 ijerph-19-02006-t006:** Sensitivity analyses of different measures of parents’ health and offspring education.

Variables	OLS	2SLS	OLS	2SLS	OLS	2SLS
	Subjective Life Expectancy		Least-Educated Children		Average Educational Attainment of All Offspring	
Panel A	All					
Children’s education	0.012 ***	0.091 ***	−0.008 ***	−0.014 ***	−0.008 ***	−0.015 ***
	(0.002)	(0.012)	(0.001)	(0.002)	(0.001)	(0.003)
Observations	6098	6098	7118	7118	7118	7118
*p*-value of Kleibergen–Paap rk LM statistic		0.000		0.000		0.000
Kleibergen–Paap rk Wald F statistic of weak identification test		178.187		161.110		232.759
*p*-value of Anderson–Rubin Wald test		0.000		0.000		0.000
*p*-value of Hansen J statistic		0.000		0.000		0.000
*p*-value of Endogeneity test		0.000		0.009		0.005
Panel B	Mothers					
Children’s education	0.011 ***	0.069 ***	−0.010 ***	−0.021 ***	−0.010 ***	−0.018 ***
	(0.003)	(0.013)	(0.001)	(0.004)	(0.001)	(0.004)
Observations	3111	3111	3727	3727	3727	3727
*p*-value of Kleibergen–Paap rk LM statistic		0.000		0.000		0.000
Kleibergen–Paap rk Wald F statistic of weak identification test		85.211		91.193		95.588
*p*-value of Anderson–Rubin Wald test		0.000		0.000		0.000
*p*-value of Hansen J statistic		0.000		0.000		0.000
*p*-value of Endogeneity test		0.000		0.007		0.012
Panel C	Fathers					
Children’s education	0.014 ***	0.118 ***	−0.005 ***	−0.006 **	−0.006 ***	−0.009 ***
	(0.004)	(0.017)	(0.001)	(0.003)	(0.001)	(0.004)
Observations	2987	2987	3391	3391	3391	3391
*p*-value of Kleibergen–Paap rk LM statistic		0.000		0.000		0.000
Kleibergen–Paap rk Wald F statistic of weak identification test		120.181		68.245		105.575
*p*-value of Anderson–Rubin Wald test		0.000		0.107		0.012
*p*-value of Hansen J statistic		0.000		0.000		0.000
*p*-value of Endogeneity test		0.000		0.774		0.272

Notes: The control variables are shown in Table 2. Standard errors clustered at the province level. *** *p* < 0.01, ** *p* < 0.05.

**Table 7 ijerph-19-02006-t007:** Placebo test results based on the year the counterfactual policy took effect.

Variables	Sample of 14–17 Years Old	Sample of 18–21 Years Old
The first stage estimation of 2SLS	0.353 (0.249)	0.013 (0.231)
*p*-value of Kleibergen–Paap rk LM statistic	0.124	0.974
Kleibergen–Paap rk Wald F statistic of weak identification test	2.164	0.001
*p*-value of Anderson–Rubin Wald test	0.455	0.551
*p*-value of Hansen J statistic	0.000	0.000
*p*-value of Endogeneity test	0.351	0.557
The second stage estimation of 2SLS	0.023 (0.033)	−0.258 (4.631)

Notes: The control variables are shown in Table 2. Standard errors clustered at the province level.

**Table 8 ijerph-19-02006-t008:** Robustness to specification and bandwidths.

Variables	OLS	2SLS	OLS	2SLS	OLS	2SLS
	[−15, 15]		[−10, 10]		[50, 70]	
Panel A	All					
Children’s education	−0.006 ***	−0.016 ***	−0.006 ***	−0.019 ***	−0.005 ***	−0.012 ***
	(0.001)	(0.003)	(0.001)	(0.004)	(0.001)	(0.003)
Observations	6727	6727	5568	5568	5307	5307
*p*-value of Kleibergen–Paap rk LM statistic		0.000		0.000		0.000
Kleibergen–Paap rk Wald<break/>F statistic of weak identification test		185.793		105.112		113.890
*p*-value of Anderson–Rubin Wald test		0.000		0.000		0.000
*p*-value of Hansen J statistic		0.000		0.000		0.000
*p*-value of Endogeneity test		0.006		0.006		0.069
Panel B	Mothers					
Children’s education	−0.006 ***	−0.021 ***	−0.006 ***	−0.026 ***	−0.005 ***	−0.017 ***
	(0.001)	(0.005)	(0.001)	(0.006)	(0.001)	(0.005)
Observations	3469	3469	2875	2875	2861	2861
*p*-value of Kleibergen–Paap rk LM statistic		0.000		0.000		0.000
Kleibergen–Paap rk Wald<break/>F statistic of weak identification test		91.212		52.497		67.902
*p*-value of Anderson–Rubin Wald test		0.000		0.000		0.001
*p*-value of Hansen J statistic		0.000		0.000		0.000
*p*-value of Endogeneity test		0.010		0.011		0.047
Panel C	Fathers					
Children’s education	−0.006 ***	−0.010 ***	−0.006 ***	−0.012 **	−0.005 ***	−0.007 *
	(0.001)	(0.004)	(0.001)	(0.005)	(0.001)	(0.004)
Observations	3258	3258	2693	2693	2446	2446
*p*-value of Kleibergen–Paap rk LM statistic		0.000		0.000		0.000
Kleibergen–Paap rk Wald<break/>F statistic of weak identification test		125.194		114.196		84.437
*p*-value of Anderson–Rubin Wald test		0.018		0.022		0.197
*p*-value of Hansen J statistic		0.000		0.000		0.000
*p*-value of Endogeneity test		0.372		0.279		0.902

Notes: The control variables are shown in Table 2. Standard errors clustered at the province level. *** *p* < 0.01, ** *p* < 0.05, * *p* < 0.1.

## Data Availability

Publicly available datasets were analyzed in this study. The datasets can be found here: http://charls.pku.edu.cn/index/en.html (accessed on 24 September 2020).

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
