# Peer review of "Offspring Education and Parents’ Health Inequality in China: Evidence from Spillovers of Education Reform"

_ijerph, 2022, doi:10.3390/ijerph19042006_

Round 1
Reviewer 1 Report
I believe the authors made a serious and thorough effort to shed light on the connection between offspring education and parental health in Chinese households. However, I would like to suggest a few recommendations:
- As the authors pointed out, they only consider one child in their study, while the analyzed families have often multiple children, and there is no information as to which of them are really the caregivers. In section 7 they refer to additional results in which they assumed that the child with the lowest educational level is taking care of the parents. In this respect, I wonder if they could gather additional information from CHARLS (i.e. their database), and compute an aggregate index of the education levels of all siblings. In my opinion, that is more appropriate and intuitive than relying on extreme assumptions.
- The authors preferred to include household's net current assets rather than income among their control variables (see page 8). Here I suggest to include both regressors, since income is often a better indicator of liquidity, although the authors argue otherwise. There may be high income households with hardly any assets, especially if the household was initially disadvantaged but the children recently found a good job, for instance. The authors should rule out the reader's suspicion that income is what matters after all.
- The authors report the OLS, the IV and the IVQR estimates automatically. However, the reader may wonder which is the preferred specification. Perhaps they could include a Hausman test to establish firmly the endogeneity of Education, in equation 4.
- I wonder why the authors did not use panel data while resorting to a longitudinal survey. They could have found much better methods to deal with the endogeneity issues then.
- I think it may be interesting to explore the implications of offspring education to improve specific parental health problems, rather than only the impact on the Frailty Index. This might enhance the contribution of the paper to the existing literature.
Author Response
We greatly appreciate the two reviewers for their valuable suggestions and comments. These suggestions and comments have helped to improve the manuscript. Thanks to the reviewers for their efforts. According to the comments of two reviewers and decision of the editor, we have made revisions to the manuscript. In the following, we give a point-by-point response to the comments.
Point 1: As the authors pointed out, they only consider one child in their study, while the analyzed families have often multiple children, and there is no information as to which of them are really the caregivers. In section 7 they refer to additional results in which they assumed that the child with the lowest educational level is taking care of the parents. In this respect, I wonder if they could gather additional information from CHARLS (i.e. their database), and compute an aggregate index of the education levels of all siblings. In my opinion, that is more appropriate and intuitive than relying on extreme assumptions.
Response 1: We gratefully appreciate for the reviewer's valuable comment. In the original manuscript, considering the more robust and stronger association between the highest educational attainment child and parents' health, we selected the child with highest educational attainment in the main text. But as the reviewer mentioned, an aggregate index of the education levels of all siblings may be more appropriate and intuitive than extreme assumptions (maximum and minimum of offspring education). We accept the reviewer's suggestion and perform a robustness analysis using the all children's average years of education in the revised manuscript. The results have not changed, which shows that our results are robust. See Table 6 (section 7.1).
Point 2: The authors preferred to include household's net current assets rather than income among their control variables (see page 8). Here I suggest to include both regressors, since income is often a better indicator of liquidity, although the authors argue otherwise. There may be high income households with hardly any assets, especially if the household was initially disadvantaged but the children recently found a good job, for instance. The authors should rule out the reader's suspicion that income is what matters after all.
Response 2: Thank you for the reviewer's suggestion. As suggested by the reviewer, we have added income variable in our analysis. Coefficient of regression result changed a little, however the findings of the manuscript has not changed.
Point 3: The authors report the OLS, the IV and the IVQR estimates automatically. However, the reader may wonder which is the preferred specification. Perhaps they could include a Hausman test to establish firmly the endogeneity of Education, in equation 4.
Response 3: Thanks for pointing out this. We included a Hausman test in the original manuscript, but did not show it in the context. We have supplemented and revised this in the manuscript (section 6.3). The results of the Hausman test show that there is endogeneity of education (p<0.01). We therefore took the IV estimation result as the preferred specification and the OLS estimation result as reference.
Point 4: I wonder why the authors did not use panel data while resorting to a longitudinal survey. They could have found much better methods to deal with the endogeneity issues then.
Response 4: We gratefully thanks for the precious time the reviewer spent making constructive remark. Firstly, 2018 CHARLS data were the most recent nationally representative data on the health of older Chinese. Secondly, as the latest CHARLS data, the 2018 survey data has been significantly different from the previous. Some variables in 2018 data were not available in the previous surveys. In addition, our manuscript primarily analyzed the relationship between offspring education and parents' health, not focusing on the time trend of the parents' health. The analysis of the changing trend of parents' health will be focused on in the future. Therefore, the purpose of our research can be achieved through cross-sectional data. There were also examples of using cross-sectional data for analysis in previous literature, such as Does children's education matter for parents' health and cognition? Evidence from China (DOI: 10.1016/j.jhealeco.2019.06.004), Children's education and parental old-age health: Evidence from a population-based, nationally representative study in India (DOI: 10.1080/00324728.2020.1775873) and Offspring Educational Attainment and Older Parents' Cognition in Mexico (DOI: 10.1215/00703370-8931725). Finally, although some results of the manuscript are similar to those obtained in the previous literature, our study also lead to new conclusions that have not been addressed in the previous.
Point 5: I think it may be interesting to explore the implications of offspring education to improve specific parental health problems, rather than only the impact on the Frailty Index. This might enhance the contribution of the paper to the existing literature.
Response 5: Thanks for the comment. We aimed at the general health of the elderly parents. As a comprehensive index, frailty index can comprehensively reflect the elderly parents' health which is also the focus of our research. Specific health problems (such as hypertension, diabetes) may have more immediate implications for policy making, but it may not reflect the general health of the elderly parents. In the future, we will be looking at specific health problems in older age cohort.
We believed all the concerns have been addressed. Thanks to the reviewers for the positive and constructive comments and suggestions on our manuscript.
Reviewer 2 Report
The article deals with the quantification of the effect of offspring education on the health of their parents in China. In this regard, regression analysis is used in order to identify and measure the role of literacy, education, societal and demographic factors in the relation between the education of children and the health level of their parents. The topic is interesting, and the study is well organized and developed. However extensive linguistic editing is needed, since the text contains many grammatical, syntax and English style errors. As a result, minor revisions is recommended.
Author Response
We greatly appreciate the two reviewers for their valuable suggestions and comments. These suggestions and comments have helped to improve the manuscript. Thanks to the reviewers for their efforts. According to the comments of two reviewers and decision of the editor, we have made revisions to the manuscript. In the following, we give a point-by-point response to the comments.
Point 1: The article deals with the quantification of the effect of offspring education on the health of their parents in China. In this regard, regression analysis is used in order to identify and measure the role of literacy, education, societal and demographic factors in the relation between the education of children and the health level of their parents. The topic is interesting, and the study is well organized and developed. However extensive linguistic editing is needed, since the text contains many grammatical, syntax and English style errors. As a result, minor revisions is recommended.
Response 1: We gratefully appreciate for the reviewer's valuable comment. As suggested, we corrected the above grammatical errors and made an effort to correct the spelling and grammar errors and polish the whole manuscript through the English editing services provided by MDPI. The attachment is the English editing certificate of MDPI and English editing ID is english-39830.
We believed all the concerns have been addressed. Thanks to the reviewers for the positive and constructive comments and suggestions on our manuscript.

Round 2
Reviewer 1 Report
Thank you very much for your paper and revision.